

# Functional analysis of a *Phytophthora* host-translocated effector using the yeast model system

Avery C. Wilson[1,2] and William R. Morgan[1]

[1] Department of Biology, The College of Wooster, Wooster, OH, United States
[2] School of Medicine, New York Medical College, Valhalla, NY, United States

## ABSTRACT

**Background:** *Phytophthora* plant pathogens secrete effector proteins that are translocated into host plant cells during infection and collectively contribute to pathogenicity. A subset of these host-translocated effectors can be identified by the amino acid motif RXLR (arginine, any amino acid, leucine, arginine). Bioinformatics analysis has identified hundreds of putative RXLR effector genes in *Phytophthora* genomes, but the specific molecular function of most remains unknown.

**Methods:** Here we describe initial studies to investigate the use of *Saccharomyces cerevisiae* as a eukaryotic model to explore the function of *Phytophthora* RXLR effector proteins.

**Results and Conclusions:** Expression of individual RXLR effectors in yeast inhibited growth, consistent with perturbation of a highly conserved cellular process. Transcriptome analysis of yeast cells expressing the poorly characterized *P. sojae* RXLR effector Avh110 identified nearly a dozen yeast genes whose expression levels were altered greater than two-fold compared to control cells. All five of the most down-regulated yeast genes are normally induced under low phosphate conditions *via* the PHO4 transcription factor, indicating that PsAvh110 perturbs the yeast regulatory network essential for phosphate homeostasis and suggesting likely PsAvh110 targets during *P. sojae* infection of its soybean host.

## INTRODUCTION

The oomycete genus *Phytophthora* includes some of the most notorious pathogens of plants (*Kamoun et al., 2015*), including *Phytophthora infestans*, the causal agent of the 1840's Irish Potato Famine, and *Phythopthora sojae*, a related species infecting soybeans and estimated to have caused as much as $2 billion in agricultural damages per year (*Tyler, 2007*). During the infection process, these filamentous microbes produce haustoria, hyphal extensions that penetrate the digested plant cell wall and create an intimate interface between the pathogen and plant cell membranes (*Boevink et al., 2020*; *Fawke, Doumane & Schornack, 2015*). To counter the plant's defense responses, the pathogen secretes a variety of effector proteins into the extrahaustorial matrix (*Wang & Jiao, 2019*). Many *Phytophthora* effector proteins appear to be translocated into the host cell, including

Corresponding author
William R. Morgan,
wmorgan@wooster.edu

a subset distinguished by an RXLR amino acid motif (RXLR = arginine, any amino acid, leucine, arginine) following the N-terminal signal peptide (*Rehmany et al., 2005*; *Wang et al., 2017*). Genome sequencing studies have revealed that the genome of *Phytophthora* and related species contain hundreds of putative genes encoding these RXLR effectors (*Haas et al., 2009*; *McGowan & Fitzpatrick, 2017*; *Tyler et al., 2006*).

While these host-translocated effectors are presumed to work together to disarm the plant immune system (*Wang & Jiao, 2019*), the exact molecular mechanism used by many RXLR effectors remains unknown. Because few RXLR effectors exhibit similarity to previously characterized proteins, hypotheses of effector function based on similar structures have been limited (*Bozkurt et al., 2012*). Indeed, RXLR effector proteins are generally too small to function as enzymes and instead presumably interact with host proteins to disrupt cellular pathways (*Wawra et al., 2012*).

The search for the specific cellular targets of pathogen effector proteins promises to shed new light on the molecular mechanisms of infection and thereby provide valuable insights for the development of more effective pathogen control strategies (*Wawra et al., 2012*). Multiple approaches have therefore been adopted to explore the molecular function of individual pathogen effectors. Studies of cellular localization following effector expression *in planta* demonstrate that following translocation into the host cell individual RXLR effectors can accumulate in the cytosol, the nucleus, various components of the endomembrane system, or some combination of these (*Caillaud et al., 2012*; *Wang et al., 2019*), suggesting that effectors may target a diverse array of host processes including host signaling pathways in the cytosol, gene expression in the nucleus, or secretion pathways in the endomembrane system. Other *in planta* work has successfully identified the specific molecular targets of some RXLR effectors, revealing a diverse set of target pathways, including callose deposition, vesicle trafficking, numerous MAPK pathways, and RNA silencing (*Wang & Jiao, 2019*; *Whisson et al., 2016*). However, the specific cellular targets of the vast majority of RXLR effectors remain unknown.

While *in planta* studies can yield much insight into the function of individual effectors, the interpretation of such studies is often complicated by the presence of plant immune receptors that recognize effectors and trigger the plant hypersensitive response, thereby masking the specific molecular action of the effector (*Munkvold et al., 2008*). The eukaryotic model organism *Saccharomyces cerevisiae* ("budding yeast"), with its powerful functional genomic resources including a comprehensive mutant collection and a rich database of extensive gene annotations (*Cherry et al., 2012*; *Giaever & Nislow, 2014*; *Norman & Kumar, 2016*), has provided an alternative approach for exploring effector function in the absence of the plant hypersensitive response (*Popa et al., 2016*; *Siggers & Lesser, 2008*). Consistent with the hypothesis that effectors often target and disrupt basic cellular processes conserved across eukaryotes, heterologous expression of many pathogen effectors in yeast significantly inhibits growth (*Munkvold et al., 2008*; *Slagowski et al., 2008*). More importantly, subsequent studies have confirmed that the molecular function of individual effectors elucidated in the yeast model system is consistently analogous to the activity in the natural host (*Popa et al., 2016*; *Siggers & Lesser, 2008*). In a pioneering study using the yeast model system to discern effector functions, *Kramer et al.*

(2007) used a combination of transcriptome analysis and high-throughput synthetic lethal studies to hypothesize that the *Shigella* effector OspF inhibits a MAPK-signaling pathway. Testing in yeast and cultured human cells confirmed this hypothesized function, and subsequent work with a mouse model demonstrated that OspF inhibition of MAPK signaling attenuates the host immune response during *Shigella* infection. Together, these findings indicate that many pathogen effectors target cellular processes conserved across eukaryotes and that yeast can be used to study how individual effectors disrupt these conserved pathways.

The investigations reported here demonstrate that heterologous overexpression of individual *P. sojae* RXLR effector genes likewise generally inhibits yeast growth. In addition, potential cellular targets of the poorly characterized *P. sojae* RXLR effector Avh110 were investigated by characterizing the genome-wide transcriptional response of *S. cerevisiae* to heterologous expression of this effector. RNA-seq analysis indicates that expression of the PsAvh110 effector significantly alters expression levels of hundreds of yeast genes. Gene ontology enrichment analysis found that the genes most repressed by PsAvh110 expression are regulated by the PHO4 transcription factor in response to phosphate conditions, identifying the phosphate homeostasis regulatory network as a likely yeast target of this RXLR effector.

## MATERIALS & METHODS

### Expression constructs

A collection of *P. sojae* (isolate P6497) RXLR effector genes (*Wang et al., 2011*) cloned in pDONR207 was received as a gift from Dr. John McDowell at Virginia Polytechnic Institute and State University. Each effector open reading frame began with the codon immediately downstream of the signal peptide cleavage site predicted by SIGNALP (*Anderson et al., 2012*). Each RXLR effector gene and the ΔGFP control gene (*Bos et al., 2006*) was individually transferred into the destination vector pAG425GAL-ccdB (*Alberti, Gitler & Lindquist, 2007*) using Gateway LR recombination reactions [Invitrogen]. LR reaction products were transformed into *E. coli* (DH5a) by electroporation (BTX Division, Genetronics, 2001) and plated onto selective media (LB+Ampicillin) using standard procedures (*Sambrook & Russell, 2001*). Following plasmid purification, pAG425GAL expression clones were transformed into *S. cerevisiae* BY4741 using a high-efficiency lithium acetate/polyethylene glycol procedure (*Gietz, 2014*).

### Yeast growth inhibition assay

Media preparation and routine manipulations for working with *S. cerevisiae* were performed according to standard procedures (*Lundblad & Struhl, 2008*). Yeast transformants were inoculated into tubes containing 3 mL of non-inducing selective media supplemented with 2% glucose (SC-LEU + 2% Glu). Following overnight growth with aeration, each saturated culture (10 μL) was diluted into non-inducing selective media supplemented with 2% raffinose (190 μL SC-LEU + 2% Raf) in a 96-well plate and incubated until $OD_{595}$ measured between 0.3–0.4 using a Multiskan FC microplate photometer. Each raffinose culture (2 μL) was then diluted into inducing selective media

containing 2% galactose (198 µL SC-LEU + 2%Gal). Following induction, the optical density of each sample was periodically measured until the $OD_{595}$ reached 1.0. All incubations were performed with rapid shaking at 30 °C.

Statistical procedures were performed in R version 3.6.3 (*R Core Team, 2020*) (Article S1). The growth rate (number of doublings per hour) of each yeast culture was calculated by log transforming the $OD_{595}$ values at each time point (hours post-induction) and using the least squares method to calculate *b*, the slope of the best fit line. Replicates with measurements poorly fitting the linear model ($r^2 < 0.95$) were excluded from further analysis. The growth rate was normalized by dividing *b* of each experimental culture by the mean value of *b* of control (ΔGFP) cultures grown at the same time. Planned comparison *t*-tests were performed as described in (*Hothorn, Bretz & Westfall, 2008*).

## Sample preparation and RNA-sequencing

RNA was prepared from triplicate cultures of yeast cells expressing *PsAvh110* or *ΔGFP*, as follows. Saturated cultures of transformed yeast grown in non-inducing selective media (SC-LEU + 2% Glu) were diluted in SC-LEU + 2% Raf to an $OD_{595}$ of 0.3 and then incubated with aeration at 30 °C. After 2 h, galactose was added to a 2% final concentration to induce gene expression from the GAL1 promoter. Two hours after induction, 1.5 mL of each culture was collected by centrifugation at $6,000 \times g$ for 3 min, and the cell pellet was frozen at −80 °C. Total RNA was isolated from each sample using the MasterPure Yeast RNA Purification Kit (Epicentre) with the optional DNA treatment procedure. RNA quality was assessed with the BioAnalyzer RNA6000 (Agilent Technologies) and passed quality control. RNA TruSeq library preparation and Illumina sequencing (75 bp single-end reads) were performed at Applied Biological Materials using standard Illumina protocols. Average insert size ranged from 330 to 411 bp and yielded 12 to 17 million reads per library.

## Bioinformatics analyses

After uploading the sequencing read files to the usegalaxy.org public server, read processing and mapping were performed on the Galaxy web platform (*Afgan et al., 2018*), as follows. Low quality regions were trimmed from each read using Trimmomatic (*Bolger, Lohse & Usadel, 2014*), and then processed reads were mapped to the sacCer3 *S. cerevisiae* reference genome using HISAT2 (*Kim, Langmead & Salzberg, 2015*). After counting the number of reads mapping to each yeast gene using featureCounts (*Liao, Smyth & Shi, 2014*), genes differentially expressed between the experimental (PsAvh110) and control (ΔGFP) samples were identified using DESeq2 (*Love, Huber & Anders, 2014*). Genes with significant expression differences (adjusted *p*-value < 0.05) were divided into up- and down-regulated genes and ordered by log fold change. Gene set enrichment analysis was performed with each ordered list of differentially expressed genes using g: Profiler (version e100_eg47_p14_7733820) with the g:SCS multiple testing correction method applying a significance threshold of 0.05 (*Raudvere et al., 2019*).

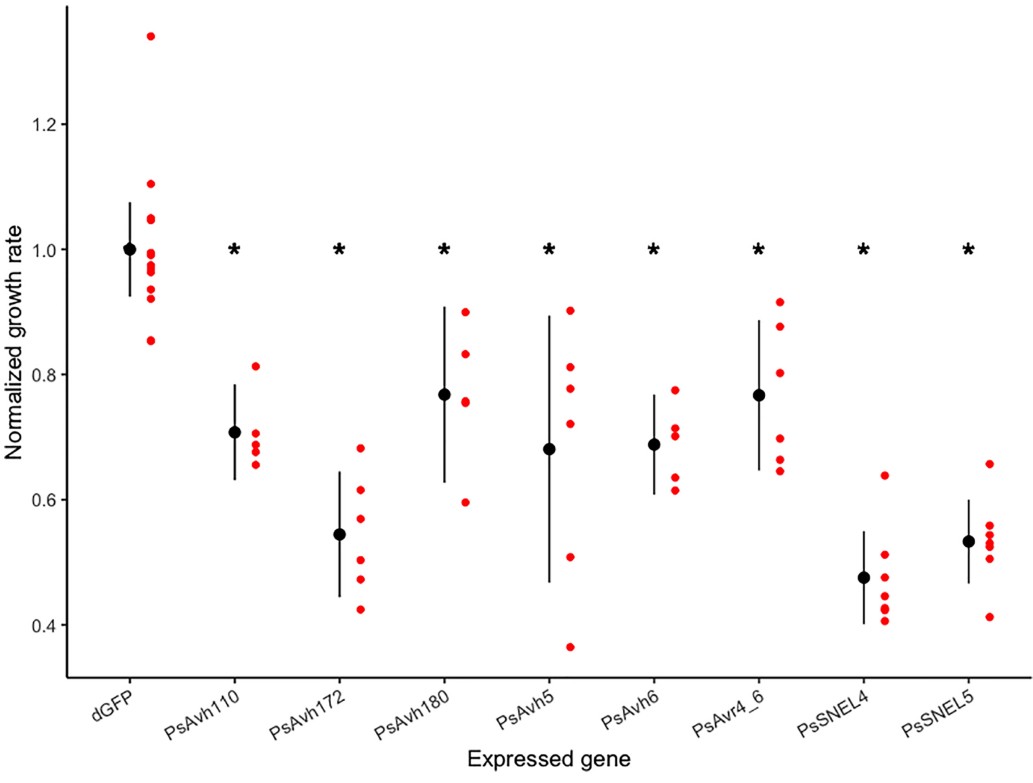

**Figure 1 Normalized growth rates of yeast strains expressing individual *P. sojae* RXLR effectors.** Each red dot indicates the normalized growth rate (relative to the control mean) of a yeast culture; each black dot and vertical line indicates the mean and 95% confidence interval of a transformed strain. Asterisks indicate strains with significantly reduced mean growth rates (planned comparison *t*-test, adjusted *p*-values < 0.01).

# RESULTS

## Overexpression of *P. sojae* RXLR effectors inhibit yeast growth

To screen for *P. sojae* RXLR effectors that inhibit yeast growth, we began with a collection of 24 *P. sojae* RXLR genes previously shown to suppress PAMP- and effector-triggered programmed cell death *in planta* (*Wang et al., 2011*). Yeast expression constructs with six of these genes were successfully transformed into *S. cerevisiae*, and the growth rate of the transformed strains was compared to a control strain expressing a negative control gene (ΔGFP; Fig. 1). Expression of each effector tested in yeast significantly inhibited growth (planned comparison *t*-tests, adj. *p*-values < 0.01), reducing the normalized growth rate from 20% to 50% (Fig. 1, Article S1). Of the most growth inhibitory RXLR effectors, *Wang et al. (2011)* previously demonstrated that PsAvh172 suppresses a MAPK signaling pathway, so we focused our attention on the other effectors, arbitrarily selecting PsAvh110, whose molecular mechanism of cell death suppression was unknown, for this pilot study.

## PsAvh110 represses many PHO4-regulated yeast genes

In this study, we investigated the yeast cellular processes targeted by the poorly characterized RXLR effector PsAvh110 by performing a comparative transcriptome

**Table 1 Most up-regulated yeast genes following PsAvh110 expression.**

| LFC* | Gene name | Protein description & function (SGD) |
|---|---|---|
| 1.5 | HXT3 (YDR345C) | Low affinity glucose transporter of the major facilitator superfamily; expression is induced in low and high glucose conditions |
| 1.3 | FRM2 (YCL026C-A) | Type II nitroreductase; possible role in lipid signaling, oxidative stress response, and metal stress response |
| 1.2 | ATO3 (YDR384C) | Putative ammonium transporter; possible role in export of ammonia from the cell |
| 1.1 | HXT4 (YHR092C) | High-affinity glucose transporter of the major facilitator superfamily, expression is induced by low levels of glucose and repressed by high levels of glucose |
| 1.0 | NFG1 (YLR042C) | Cell wall protein of unknown function; deletion improves xylose fermentation in industrially engineered strains |
| 1.0 | (YMR244W) | Putative protein of unknown function |

Note:
* Genes differentially expressed between the experimental (PsAvh110) and control (ΔGFP) samples were identified using DESeq2 and ordered by log fold change as described in the Materials and Methods section. LFC, $Log_2$ fold-change; Standard (& systematic) gene names; SGD, *Saccharomyces* Genome Database (*Cherry et al., 2012*).

**Table 2 Most down-regulated yeast genes following PsAvh110 expression.**

| LFC* | Gene name | Protein description & function (SGD) |
|---|---|---|
| −2.3 | PHO89 (YBR296C) | Sodium: inorganic phosphate symporter involved in phosphate ion transmembrane transport |
| −2.1 | SPL2 (YHR136C) | Similarity to cyclin-dependent kinase inhibitors; downregulates low-affinity phosphate transport during phosphate limitation |
| −1.5 | PHO84 (YML123C) | High-affinity inorganic phosphate (Pi) transporter involved in phosphate ion transport and polyphosphate metabolism |
| −1.1 | PHO5 (YBR093C) | One of three repressible acid phosphatases; involved in the cellular response to phosphate starvation and the metabolism of phosphate-containing compounds |
| −1.1 | PHO12 (YHR215W) | One of three repressible acid phosphatases; regulated by phosphate starvation |

Note:
* Genes differentially expressed between the experimental (PsAvh110) and control (ΔGFP) samples were identified using DESeq2 and ordered by log fold change as described in the Materials and Methods section. LFC, $Log_2$ fold-change; Standard (& systematic) gene names; SGD, *Saccharomyces* Genome Database (*Cherry et al., 2012*).

analysis. To identify yeast genes that were up- or down-regulated in response to the PsAvh110 effector, RNA sequencing reads were generated from triplicate cultures of transformed yeast strains expressing either the PsAvh110 effector gene or a negative control gene (ΔGFP). After mapping reads to the *P. sojae* reference genome, individual transcript levels in each sample were calculated, and computational analyses were conducted to identify genes that were differentially expressed in the presence of the RXLR effector. Subsequently, gene set enrichment analysis was used to identify features over-represented in sets of differentially expressed genes.

Of the 6,445 detected yeast genes, 851 (13.2%) had significantly different expression levels in cells expressing PsAvh110 as compared to negative control cells (adjusted $p$-value < 0.05). Of these, 488 genes were significantly up-regulated (six > 2-fold), and 363 genes were significantly down-regulated (five > 2-fold). Three of the six genes up-regulated >2-fold ($log_2$-fold change (lfc) > 1) encode monovalent cation transporters and contain gene regulatory regions including a Mig1p binding motif (Table 1; Table 3). The five genes down-regulated >2-fold (lfc < −1) all have putative roles in phosphate homeostasis (Table 2). Two of these genes encode presumed inorganic phosphate transporters (PHO89, PHO84), two encode repressible acid phosphatases (PHO5 and PHO12), and the fifth

**Table 3 Gene set enrichment analysis of most up-regulated yeast genes.**

| TERM ID | Term name | Score* | Term size | Query size | Intersecting genes |
|---|---|---|---|---|---|
| GO:0005353 | Fructose transmembrane transporter activity | 2.43 | 15 | 4 | YDR345C, YHR092C |
| GO:0015578 | Mannose transmembrane transporter activity | 2.43 | 15 | 4 | YDR345C, YHR092C |
| TF:M00061_0 | Factor: Mig1p; motif: KANWWWWATSYGGGGWA; match class: 0 | 2.83 | 74 | 6 | YDR345C, YDR384C, YHR092C |

Notes:
Gene set enrichment analysis using g: Profiler (see Material and Methods) was performed with the ordered list of genes with $\log_2$-fold change > 1 as the query. Terms with a score > 2 are shown. Data sources included all three Gene Ontology (GO) categories, KEGG biological pathways (KEGG), and TRANSFAC regulatory motifs in DNA (TF).
* Score equals $-\log_{10}$ (adj. $p$-value); term size and query size refer to the number of genes in each list; intersecting genes are common to the term and query gene lists.

**Table 4 Gene set enrichment analysis of most down-regulated yeast genes.**

| TERM ID | Term name | Score* | Term size | Query size | Intersecting genes |
|---|---|---|---|---|---|
| GO:0005315 | Inorganic phosphate transmembrane transporter activity | 3.10 | 7 | 3 | YBR296C, YML123C |
| GO:0003993 | Acid phosphatase activity | 2.45 | 8 | 5 | YBR093C, YHR215W |
| GO:0006817 | Phosphate ion transport | 2.03 | 13 | 3 | YBR296C, YML123C |
| KEGG:00740 | Riboflavin metabolism | 2.51 | 14 | 5 | YBR093C, YHR215W |
| KEGG:00730 | Thiamine metabolism | 2.24 | 19 | 5 | YBR093C, YHR215W |
| TF:M00064_0 | Factor: Pho4p; motif: TNVCACGTKGGN; match class: 0 | 7.33 | 100 | 5 | YBR296C, YHR136C, YML123C, YBR093C, YHR215W |
| TF:M00064_1 | Factor: Pho4p; motif: TNVCACGTKGGN; match class: 1 | 5.21 | 15 | 5 | YHR136C, YBR093C, YHR215W |
| TF:M01564_1 | Factor: Pho4p; motif: NNNNNNSCACGTGSNNNNNN; match class: 1 | 3.75 | 93 | 3 | YBR296C, YHR136C, YML123C |
| TF:M01564_0 | Factor: Pho4p; motif: NNNNNNSCACGTGSNNNNNN; match class: 0 | 3.50 | 113 | 3 | YBR296C, YHR136C, YML123C |
| TF:M01699_1 | Factor: Cbf1p; motif: CACGTG; match class: 1 | 3.08 | 696 | 5 | YBR296C, YHR136C, YML123C, YBR093C, YHR215W |
| TF:M01699_0 | Factor: Cbf1p; motif: CACGTG; match class: 0 | 3.08 | 696 | 5 | YBR296C, YHR136C, YML123C, YBR093C, YHR215W |

Notes:
Gene set enrichment analysis using g: Profiler (see Material and Methods) was performed with the ordered list of genes with $\log_2$-fold change < −1 as the query. Terms with a score > 2 are shown. Data sources included all three Gene Ontology (GO) categories, KEGG biological pathways (KEGG), and TRANSFAC regulatory motifs in DNA (TF).
* Score equals $-\log_{10}$ (adj. $p$-value); term size and query size refer to the number of genes in each list; intersecting genes are common to the term and query gene lists.

encodes a CDK inhibitor homolog (SPL2) that regulates phosphate homeostasis. The regulatory region of all five genes includes a Pho4p binding motif (Table 4).

# DISCUSSION

## *P. sojae* RXLR effectors generally target conserved eukaryotic cellular processes

*Saccharomyces cerevisiae* has been utilized in numerous studies for the investigation of host-translocated effectors of bacterial pathogens based on the observation that these

effectors often target fundamental cellular processes conserved among eukaryotes (*Popa et al., 2016*; *Siggers & Lesser, 2008*). This study examined the efficacy of yeast as a model for the functional characterization of *Phytophora* RXLR effectors by first examining if heterologous expression of individual effectors inhibited yeast growth, indicating disruption of a critical cellular process (*Munkvold et al., 2008*; *Slagowski et al., 2008*). By monitoring yeast growth using a quantitative micro-scale assay, expression of all eight *P. sojae* RXLR effectors tested was found to significantly inhibit yeast growth (Fig. 1) compared to a negative control (ΔGFP) transformant. We conclude that *Phytophthora* host-translocated effectors often perturb conserved cellular processes and that yeast can be a useful model system for exploring their molecular function.

## The PsAvh110 effector may disrupt PHO4 regulation

While the growth inhibition results suggested that each *Phytopthora* RXLR effector targets a conserved eukaryotic cellular process, such studies do not yield insights into the particular molecular targets of individual effectors. To generate hypotheses for the virulence function of a specific *P. sojae* effector, we examined the yeast transcriptome response to heterologous expression of the poorly characterized *P. sojae* Avh110 effector, which inhibited the yeast growth rate by approximately 30% (Fig. 1). By identifying shared functional themes in the yeast genes differentially expressed in response to PsAvh110 expression, we hoped to generate testable hypotheses for the molecular function of this effector in yeast and potentially in the natural plant host (*Kramer et al., 2007*).

Among the six genes up-regulated two-fold or more in the PsAvh110-expressing strain, the only commonality is that three have a regulatory region with binding sites for the glucose-regulated Mig-1 transcription factor (Table 3), including two that encode hexose transporters (Table 1). However, the weaker evidence associated with these patterns ($p$-values $> 1 \times 10^{-3}$) indicates that many hexose transporter and Mig-1-regulated genes were not among the most up-regulated genes. In contrast, a more striking pattern emerges among the yeast genes that were down-regulated two-fold or more (Table 2). All five of these most down-regulated genes are up-regulated in a PHO4-dependent manner in low phosphate conditions (*Ogawa, DeRisi & Brown, 2000*) and function during phosphate starvation: The secreted acid phosphatases PHO5 and PHO12 free inorganic phosphate from extracellular substrates, which the high affinity $P_i$ transporters PHO84 and PHO89 then transport into the cell; meanwhile, SPL2 down-regulates the low affinity phosphate transporters (*Wykoff et al., 2007*).

The regulatory regions of all five of the most down-regulated genes possess binding sites for the PHO4 transcription factor (Table 4). In response to low phosphate conditions, PHO4 is activated and significantly induces the expression of roughly 20 genes (*Ogawa, DeRisi & Brown, 2000*). Given that our yeast cultures were grown in replete phosphate conditions, where the activation of the PHO-regulated genes is not expected, it seemed paradoxical to detect any difference in PHO gene expression upon PsAvh110 expression. However, comparable reductions in PHO-regulated gene expression have previously

been seen in *pho4* deletion mutants (*Chua et al., 2006*; *He, Zhou & O'Shea, 2017*) (Article S2), supporting the hypothesis that PsAvh110 expression either directly or indirectly inhibits PHO4 activity.

Activity of the yeast PHO4 transcription factor is post-transcriptionally regulated at the level of both mRNA stability and protein phosphorylation. PHO92 facilitates the degradation of PHO4 mRNA by binding the 3′ untranslated region in association with POP2 (*Kang et al., 2014*). Given that PHO4 mRNA levels do not differ significantly in PsAvh110-expressing yeast cells (Article S2), stimulation of PHO92 activity is unlikely to produce the observed reductions in PHO regulon mRNAs. Rather, PsAvh110 apparently targets the well-conserved signaling pathway that regulates PHO4 activity *via* phosphorylation (*Ogawa, DeRisi & Brown, 2000*). In replete phosphate conditions, the PHO80-PHO85 cyclin-cyclin dependent kinase (CDK) complex phosphorylates PHO4, redirecting it to the cytoplasm. The PHO80-PHO85 complex in turn is negatively regulated by the CDK inhibitor PHO81 in association with the small metabolite inositol heptakisphosphate (IP7), a product of the inositol hexakisphosphate kinase (IP6K) VIP1 (*Lee et al., 2007*; *Secco et al., 2012*). Many of these proteins critical to phosphate homeostasis are widely conserved among eukaryotes, including plants (*Secco et al., 2012*), and could potentially be the target of the PsAvh110 effector.

## Host-translocated effectors consistently inhibit yeast growth

The observation that all eight *P. sojae* host-translocated effectors tested inhibited yeast growth initially raised concern that this effect might be due to a non-specific artifact of protein overexpression in yeast. However, several lines of evidence argue against this. First, most genes when overexpressed in the yeast system do not inhibit growth. Most notably, *Liu, Krizek & Bretscher (1992)* found that overexpression of <1% of yeast genes caused severe growth inhibition in yeast. Similarly, *Slagowski et al. (2008)* observed that few of the 20 non-translocated *Shigella* proteins had this effect. In contrast, all 19 *Shigella* host-translocated effectors tested did significantly inhibit yeast growth. Finally, a separate RNA-seq study conducted in our lab found that the transcriptional response elicited by a second *P. sojae* effector, PsAvh172, had a profile clearly distinct from the transcriptional response described here (M Reeder, W Morgan, 2020, unpublished data; EMBL-EBI ArrayExpress E-MTAB-4682). Taken together, these multiple lines of evidence argue that the growth inhibition elicited by each pathogen effector is due to its specific activity in the yeast cell and not a general response to protein overexpression.

## Future work

Further work is required to identify the precise molecular mechanism of PsAvh110 action in yeast and ultimately the plant host. Immediate steps are to examine the phosphorylation state and subcellular localization of PHO4 in yeast cells expressing PsAvh110 grown under low phosphate conditions (*Springer et al., 2003*). Once confirmed, immunoprecipitation and the yeast two-hybrid system can be used to test for molecular interactions between components of the PHO signaling pathway and PsAvh110.

Although this proposed function in regulating nutrient acquisition was unanticipated given Avh110's reported function in suppression of the plant immune response, recent work has demonstrated that alterations in the cellular phosphate status of plants leads to downstream molecular events that converge on the plant immune response, including synthesis of phytohormones, induction of phosphate stress response genes, and the redistribution of phosphate transporters (*Castrillo et al., 2017*; *Chan, Liao & Chious, 2020*; *Wang et al., 2011*). Indeed, a recent study demonstrated that altering phosphate conditions induced a change in the relationship between *Arabidopsis thaliana* and certain microorganisms, from either neutral to beneficial or beneficial to pathogenic depending on the microorganism (*Morcillo et al., 2020*). These discoveries suggest a plausible mechanism by which Avh110 can alter the immune response of the plant by influencing pathways involved in phosphate acquisition. Future studies *in planta* can further elucidate this function.

## CONCLUSIONS

The current study demonstrates the power of yeast functional genomics in generating working hypotheses regarding the molecular function of host-translocated *Phytophthora* effectors. Expression of individual RXLR effectors in yeast cells inhibited growth, consistent with the premise that these pathogen proteins often target well conserved cellular processes. Transcriptome analysis of the yeast response to the *P. sojae* Avh110 effector expression suggests that regulation of PHO4 transcription factor activity, which mediates the yeast response to limiting phosphate conditions, is a major target of this RXLR effector. Further work is needed to identify which component of the PHO4 regulatory network is the precise target of PsAvh110 in yeast and to then elucidate if an orthologous protein in plants is similarly targeted. The yeast-based model for investigating the function of *Phytophthora* effectors promises to increase our molecular understanding of the pathogenicity mechanisms of these devastating plant pathogens with the ultimate goal of developing more robust strategies for pathogen management.

## ACKNOWLEDGEMENTS

We acknowledge the superior technical work of Matthew Reeder who conducted the yeast growth inhibition assays. We thank Beth Lingenfelter for her administrative assistance.

### Funding

This research was funded by the Agriculture and Food Research Initiative competitive grant 2011-68004-30104 of the USDA's National Institute of Food and Agriculture, as well as The Henry J. Copeland Fund for Independent Study and The Theron L. Peterson and Dorothy R. Peterson Biology Research and Expense Fund at The College of Wooster. The funders had no role in study design, data collection and analysis, decision to publish, or preparation of the manuscript.

## Grant Disclosures

The following grant information was disclosed by the authors:
Agriculture and Food Research Initiative competitive grant: 2011-68004-30104.
USDA's National Institute of Food and Agriculture.
Independent Study.
College of Wooster.

## Competing Interests

The authors declare that they have no competing interests.

## Author Contributions

- Avery C. Wilson conceived and designed the experiments, performed the experiments, analyzed the data, authored or reviewed drafts of the paper, and approved the final draft.
- William R. Morgan conceived and designed the experiments, analyzed the data, prepared figures and/or tables, authored or reviewed drafts of the paper, and approved the final draft.

## DNA Deposition

The following information was supplied regarding the deposition of DNA sequences:
RNA-seq data are available in the EMBL-EBI ArrayExpress database: E-MTAB-9566.
https://www.ebi.ac.uk/arrayexpress/experiments/E-MTAB-9566/.

## Data Availability

The raw yeast growth data (optical density at 595 nm), the code for its analysis in the R notebook file, and the code for the analysis of matching yeast transcriptome profiles in the R notebook are available in the Supplemental File.
The Galaxy history of the transcriptome data analysis is accessible at https://usegalaxy.org/u/wmorgan/h/psavh110-rna-seq-analysis.

## Supplemental Information

Supplemental information for this article can be found online at http://dx.doi.org/10.7717/peerj.12576#supplemental-information.

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
