# Peer review of "Functional analysis of a Phytophthora host-translocated effector using the yeast model system"

_PeerJ, doi:10.7717/peerj.12576_

## Round 0.1 · original submission · Major Revisions

Both reviewers think that the article was interesting, but there were some issues that needed to be refined and revised. Please make careful changes in accordance with their opinions. Especially the description of the figures and tables.

Reviewer 1 ·

Basic reporting

no comment

Experimental design

Question1: At least 3 P. sojae effectors severely inhibited yeast growth compared to Avh110,why selected Avh110? authors should make some statements to explain that.

Question2: Please confirm the expression of genes listed in Table 1 and Table 2 by RT-qPCR in yeast.

Question3: It is rational to test whether the expression pattern of interesting genes in host plant is similar to the orthologous in yeast to support authors hypothesis "pathogen effectors target cellular processes conserved across eukaryotes and that yeast can be used to study how individual effectors disrupt these conserved pathways."

Validity of the findings

The evidence of this research is not sufficient enough to support the validity of functional analysis of a Phytophthora host-translocated effector using the yeast model system. Functional verification in plants is necessary.

Additional comments

Using the yeast model system to explore the function of Phytophthora RXLR effector proteins is interesting. Transcriptome analysis and yeast growth inhibition assay could not thoroughly elucidate the molecular mechanism of PsAvh110 action in the host plant. Authors should provide more sufficient evidence to show that the yeast system is similar to the plant system which is sutible for plant-pathogen interaction investigation.

·

Basic reporting

In general, the manuscript is well-organized and the writing is very clear. However, I have a few suggestions for improvement.

1. Add more explanation of the legend of Figure 2
2. Line 140: Add some information about standard RNAseq quality control metrics to confirm the quality of the data set.
3. The legends of Tables 1 and 2 need editing.
4. Introduction: Include a brief summary of a strong success story from use of yeast to investigate bacterial effectors, to help the reader understand the potential value of this approach.
5. Explain how Figures 2 and 3 were generated.

Experimental design

1. Line 110: Replace “dummy” with “control” and explain more about this gene. Does it encode a protein or is it really “expressing a non-functional transcript”. If the latter, is it possible that the reduced growth of this six effector-overexpressing strains could simply reflect growth reduction due to non-specific protein production rather than specific growth inhibiting effects of each effector gene? One way to disprove this hypothesis is to include data from any other RXLR-overexpressing strain in which growth inhibition was not observed.
2. Line 173: Specify how many effector genes were tested and explain why these six were selected from the hundreds of possible choices. Provide a little more info about what’s known about each gene from previous studies, either here or in the discussion.

Validity of the findings

1. Although the data are sound overall, Point 1 under experimental design must be addressed.

Additional comments

I commend the authors for undertaking this study. Yeast has been under-utilized for studies of oomycete effectors, and I hope that molecular oomycologists take note of these promising results.

I suggest some expansion of the following points in the discussion, to better contextualize these results and maximize the reader’s appreciation of the study’s impact:

1. Expand your discussion of the high “success rate” in the initial screen. Did it surprise you that all six effectors inhibit yeast growth? How did this hit rate compare with previous screens of effectors from other pathogens. Could the success rate be explained by a bias in selection for genes that had previously been associated with suppression of plant immunity?
2. Avh110 had been previously associated with suppression of plant immune responses. The proposed function of this gene would seem to relate more to nutrient acquisition. Might these functions be somehow reconciled? It might be useful to look at recent literature that associates phosphate transport/metabolism with plant immunity.
3. Although you report that the upregulated genes have no common theme, it seems striking that three of the six most highly-upregulated genes are also transporters, albeit of different nutrients. Does this hold additional implications for the AvhL110’s function? Might Mig-1 also be a target?

---

## Round 0.2 · Minor Revisions

You have answered most of the questions of the reviewers in the previous round well. We all agree that you have done an excellent job with this work. However, a new reviewer read your article carefully and made comments.

Based on the advice received, we have decided that your manuscript can be accepted for publication after you have carried out the corrections as suggested by the reviewer(s). These should be successfully addressed, otherwise not responding accordingly can still result in rejection of the paper.

In particular, the first question was raised similarly by a reviewer in the previous round, so I strongly recommend that you conduct this additional experiment or add sufficient discussion to the manuscript to convince the reviewers and readers.

Figures 2 and 3 can also be redrawn or generated as two reviewers have mentioned this issue.

Reviewer 3 ·

Basic reporting

This manuscript shows that six P. sojae effector that suppress PAMP- and effector-triggered programmed cell death in planta inhibit yeast growth, which suggests effector may perform similar molecular function in two different species. It can provide some clues about potentially key regulatory genes in plants and lays the foundation for future work thought the yeast model system that explores the function of phytophthora effector. There is no doubt that there are some differences between the yeast system and phytophthora-plant interaction system. But in the face of the large number of effectors secreted by pathogens, the yeast system also provides us with a way to quickly screen for potentially functional effectors and explore their possible molecular mechanisms providing support for studies on plant-pathogen interactions. Overall, the manuscript is well written and the results are easy to follow.

Experimental design

Several aspects of the study need further interpretation or additional experimentation.
1. These six effectors can inhibit yeast growth, which is probably caused by the experimental system. To exclude this possibility, author should add effectors that can’t suppress PAMP- and effector-triggered programmed cell death as a control to make the results more persuasive.
2. The transcriptome analysis found PHO4 may be a potential target of PsAvh110. The RT-qPCR related experiments about PHO4-regulated genes expression in yeast should be performed, which can verify whether these transcriptome analysis data are reliable to a certain extent.
3. Analysis data related software helps us analyze and provide convenience. With reliable data, we also have the responsibility to do our best to show it. So the pictures shown in Figure 2 and Figure 3 should be drawn with related drawing software or displayed in a new way instead of screenshots.

Validity of the findings

no comment

Additional comments

no comment

---

## Round 0.3 · accepted · Accept

Your revision was basically able to explain the reviewers' questions and improve the quality of the paper. So this manuscript can be accepted for publication. Congratulations!